# Machine Learning for Practical Quantum Error Mitigation

## Abstract

Quantum computers are actively competing to surpass classical supercomputers, but quantum errors remain their chief obstacle. The key to overcoming these on near-term devices has emerged through the field of quantum error mitigation, enabling improved accuracy at the cost of additional runtime. In practice, however, the success of mitigation is limited by a generally exponential overhead. Can classical machine learning address this challenge on today's quantum computers? Here, through both simulations and experiments on state-of-the-art quantum computers using up to 100 qubits, we demonstrate that machine learning for quantum error mitigation (ML-QEM) can drastically reduce overheads, maintain or even surpass the accuracy of conventional methods, and yield near noise-free results for quantum algorithms. We benchmark a variety of machine learning models—linear regression, random forests, multi-layer perceptrons, and graph neural networks—on diverse classes of quantum circuits, over increasingly complex device-noise profiles, under interpolation and extrapolation, and for small and large quantum circuits. These tests employ the popular digital zero-noise extrapolation method as an added reference. We further show how to scale ML-QEM to classically intractable quantum circuits by mimicking the results of traditional mitigation results, while significantly reducing overhead. Our results highlight the potential of classical machine learning for practical quantum computation.

## 1 Introduction

Quantum computers promise remarkable advantages over their classical counterparts, offering solutions to certain key problems with speedups ranging from polynomial to exponential Biamonte et al. [2017], Daley et al. [2022]. Despite significant progress in the field, the practical realization of this advantage is hindered by inevitable errors in the physical quantum devices. Quantum error mitigation (QEM) strategies have been developed to harness imperfect quantum computers to nonetheless yield near noise-free and meaningful results despite the presence of unmonitored errors Bravyi et al. [2022], Cai et al. [2022]. Crucially, QEM is paving the way to near-term quantum utility and a path to outperform classical supercomputers Daley et al. [2022], Kim et al. [2023a]

Quantum error mitigation as such aims to enhance the accuracy of a noisy quantum computation but at the cost of extended execution times. It has played a pivotal role in expanding the computational horizons of extant noisy quantum devices Kandala et al. [2019], Kim et al. [2023b,a] and has been instrumental in realizing preliminary manifestations of quantum advantage Daley et al. [2022], Bravyi et al. [2022], Pokharel and Lidar [2022]. Quantum error mitigation is typically achieved by implementing a class of mitigation circuits either in tandem with or as a replacement for a target quantum circuit, representing a computation. A cornerstone approach in this domain is zero-noise extrapolation (ZNE) Temme et al. [2017], Li and Benjamin [2017]. In ZNE, the 'zero-noise' (ideal) expectation value of the target circuit is discerned by extrapolating from expectation values across a spectrum of increased noise intensities. Strictly speaking the ZNE method does not guarantee unbiased data estimators, as certain other key QEM methods do, such as probabilistic error cancellation (PEC) Temme et al. [2017], Li and Benjamin [2017], van den Berg et al. [2023]. While these latter methods are fortified with rigorous theoretical guarantees and promise enhanced accuracy, they also demand exponential sampling overheads Quek et al. [2022], Takagi et al. [2022]—making them challenging to implement at scales of interest.

Submitted to NeurIPS 2023 AI for Science Workshop.

Emerging at the crossroads of quantum mechanics and statistical learning, machine learning for quantum error mitigation (ML-QEM) presents a promising avenue where statistical models are trained to derive mitigated expectation values from noisy counterparts executed on quantum computers. Could such ML-QEM methods offer valuable improvement in accuracy or runtime efficiency in practice?

In principle, a successful ML-QEM strategy would learn the effect of noise in training, thus obviating the need for additional mitigation circuits during the execution of an algorithm. Compared to conventional QEM, the algorithmic runtime would then see a potential reduction in overhead. However, quantum noise can be complex and can drift over time, and so it would have to be learned accurately and quickly. First explorations of ML-QEM ideas have shown signs of promise, even for complex noise profiles Kim et al. [2020], Czarnik et al. [2021, 2022], Bennewitz et al. [2022], Patel and Tiwari [2021], Strikis et al. [2021], but it remains unclear if ML-QEM can perform in practice in quantum computations on hardware or at scale. For instance, it is unclear whether a given ML-QEM method can be used across different device noise profiles, diverse circuit classes, and large quantum circuit volumes beyond the limits of classical simulation. To date, there has not been a systematic study comparing different traditional methods and statistical models for QEM on equal footing under practical scenarios across a variety of relevant quantum computational tasks.

In this article, we present a general framework to perform ML-QEM for higher runtime efficiency compared to other mitigation methods. Our study encompasses a broad spectrum of simple to complex machine learning models, including the previously proposed linear regression and multi-layer perceptron model. We further propose two new models, random forests and graph neural networks. We find that random forests seem to consistently perform the best. We evaluate the performance of all four models in diverse realistic scenarios. We consider a range of circuit classes (random circuits and Trotterized Ising dynamics) and increasingly complex noise models in simulations (including incoherent and coherent gate errors, readout errors, and qubit errors). Additionally, we explore the advantages of ML-QEM methods over traditional approaches in common use cases, such as generalization to unseen Pauli observables, and enhancement of variational quantum-classical tasks. Our analysis reveals that ML-QEM methods, particularly random forest, exhibit competitive performance compared to a state-of-the-art method—digital zero-noise extrapolation (ZNE)—while requiring lower overhead by a factor of at least 2 in runtime. Finally, with experiments on IBM quantum computers for quantum circuits with up to 100 qubits and two-qubit gate depth of 40 (with up to 1,980 CNOT gates), we propose a path toward scalable mitigation by mimicking traditional mitigation methods with superior runtime efficiency, which also serves as a further example of using classical machine learning on quantum data Huang et al. [2022].

Before proceeding to results, let us summarize the general workflow of ML-QEM; see Fig. 1. In an initial training step, the model training data is generated containing, at a minimum, the noisy and target expectation values of quantum circuits that should be similar to those used in testing; the training set can also be augmented to include, for instance, encoded features of the quantum circuits and quantum backend. Then, the model is trained by minimizing a loss function over the mitigated and target expectation values. A key feature of the ML-QEM model is that at runtime, the model produces mitigated expectation values from the noisy ones without the need for additional mitigation circuits, thus dramatically reducing overheads.

## 2 Results

### 2.1 Performance Comparison at Tractable Scale

First, we present a comparative analysis of several representative ML-QEM methods. As portrayed in Fig. 7 in the Methods section in App. A.1, we explore several statistical models in our study with varying complexity and methods of encoding data, namely linear regression with ordinary least squares (OLS), random forests (RF), multi-layer perceptrons (MLP), and graph neural networks (GNN). Since the relationship between the noisy expectation values and the ideal ones is non-linear in general (see App. B for more details), we emphasize the role of non-linear machine learning models, and study three non-linear models, i.e., RF, MLP, and GNN, in addition to the linear model OLS. Each of these models is described in further detail in the Methods in App. A.1. We compare these models against each other and digital gate-folding ZNE. Future studies comparing ML-QEM against methods with more rigorous theoretical guarantees, such as probabilistic error cancellation or amplification, are warranted.

We evaluate the performance of these methods for two classes of circuits: random circuits and Trotterized dynamics of the 1D Ising spin chain on small-scale simulations. These two classes of circuits bear distinct two-qubit gate arrangements, allowing us to gain knowledge about the performance of the ML-QEM on the two extremes of the spectrum in terms of circuit structures. This evaluation is done by simulations on small-scaled circuits, conveniently

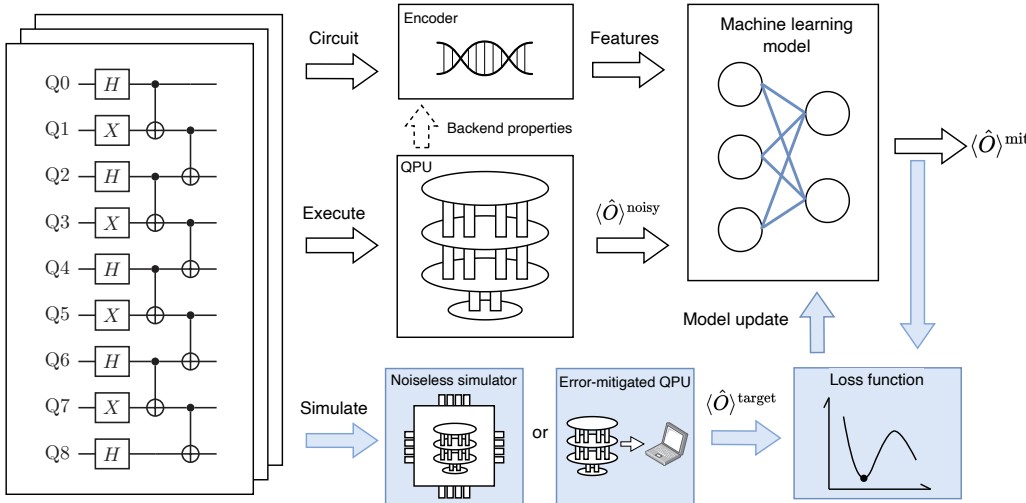

Figure 1: **Machine-learning quantum error mitigation (ML-QEM): execution and training for tractable and intractable circuits.** A quantum circuit (left) is passed to an encoder (top) that creates a feature set for the ML model (right) based on the circuit and the quantum processor unit (QPU) targeted for execution. The model and features are readily replaceable. The executed noisy expectation values $\langle \hat{O} \rangle^{\mathrm{noisy}}$ (middle) serve as the input to the model whose aim is to predict their noise-free value $\langle \hat{O} \rangle^{\mathrm{mit}}$. To achieve this, the model is trained beforehand (bottom, blue highlighted path) against target values $\langle \hat{O} \rangle^{\mathrm{target}}$ of example circuits. These are obtained either using noiseless simulations in the case of small-scale, tractable circuits or using the noisy QPU in conjunction with a conventional error mitigation strategy in the case of large-scale, intractable circuits. The training minimizes the loss function, typically the mean square error. The trained model operates without the need for additional mitigation circuits, thus reducing runtime overheads.

90 allowing us to vary the type of noises affecting the circuits and to identify situations under which the ML-QEM
91 outperforms digital ZNE in terms of mitigation accuracy.

92 ### 2.1.1 Random Circuits

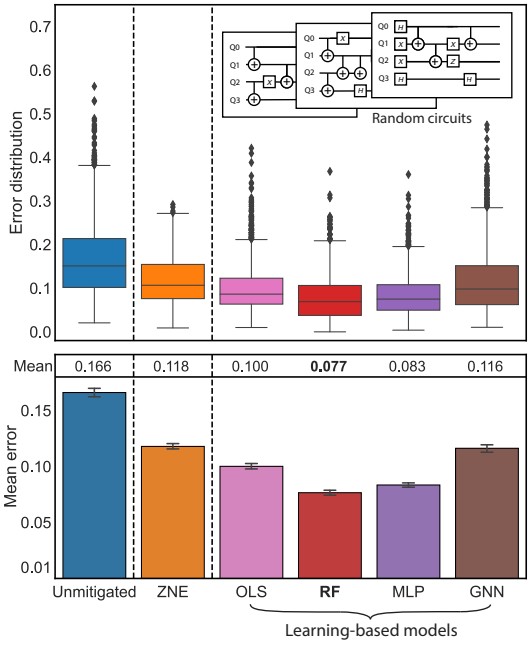

Figure 2: **Quantum error mitigation (QEM) and ML-QEM accuracy on random circuits.** Top: Error distribution for unmitigated and mitigated Pauli-Z expectation values. Mitigation is performed using either a reference QEM method, digital zero-noise extrapolation (ZNE), or one of four ML-QEM models (explained in text). Inset: Example random circuits. Noisy execution is numerically simulated using a noise model derived from IBM QPU Lima. The error is defined as the $L_2$ distance between the vector of all ideal and noisy single-qubit expectations $\langle \hat{Z}_i \rangle$; i.e., $\|\langle \hat{Z} \rangle - \langle \hat{Z} \rangle_{\mathrm{ideal}}\|_2$. Black dots are outliers. Average is over 2,000 four-qubit random circuits, with two-qubit-gate depths sampled up to 18. Bottom: Average error for each method (using data from the top) is presented with $95\%$ confidence intervals, derived from bootstrap re-sampling. The mean $L_2$ error is provided above each column.

93 In the first experiment, we benchmark the performance of the protocol on small-scale unstructured circuits. To
94 ensure that the circuits encompass a broad spectrum of complexities, we generate a diverse set of four-qubit

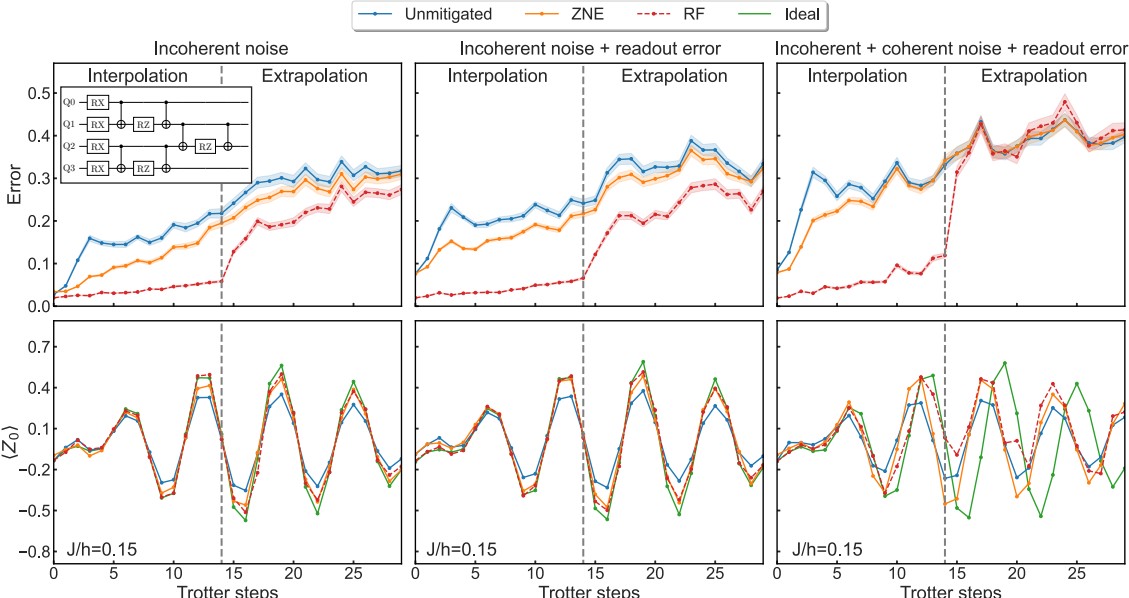

Figure 3: **Mitigation accuracy under i) complexity of quantum noise and ii) ML-QEM interpolation and extrapolation for Trotter circuits.** Top row: Average error performance on Trotter circuits (top-left inset) representing the quantum time dynamics of a four-site, 1D, transverse-field Ising model in numerical simulations. A Trotter step comprises four layers of CNOT gates (inset). Vertical dashed line separates experiments in the ML-QEM interpolation regime (left) from the extrapolation regime (right). The 3 curves represent the performance of the highest-performing ML-QEM method, the QEM ZNE method, and the unmitigated simulations. They are averaged over 300 circuits, each with a randomly chosen Pauli measurement bases. The data is for all four weight-one expectations $\langle \hat{P}_i \rangle$. The error is defined as $L_2$ distance from the ideal expectations, $\|\langle \hat{P} \rangle - \langle \hat{P} \rangle_{\text{ideal}}\|_2$, as also defined for the remainder of figures. From the left to right, the complexity of the device noise model increases to include additional realistic noise types. Coherent errors are introduced on CNOT gates. Bottom row: Corresponding typical data of the error-mitigated expectation values of the $\langle Z_0 \rangle$ Trotter evolution; here, for Ising parameter ratio $J/h = 0.15$.

random circuits with varying two-qubit gate depths, up to a maximum of 18, as shown in the inset of Fig. 2. Per two-qubit gate depth, there are 500 random training circuits and 200 random test circuits that are generated by the same sampling procedure. For each circuit, we carry out simulations on IBM's `FakeLima` backend, which emulates the incoherent noise present in the real quantum computer, the `ibmq_lima` device. While these quantum devices generally have coherent errors as well, they can be suppressed through a combination of e.g., dynamical decoupling Ezzell et al. [2022] and randomized compiling van den Berg et al. [2023], Wallman and Emerson [2016]. Specific types of noise include incoherent gate errors, qubit decoherence, and readout errors. We train the ML-QEM models to mitigate the noisy expectation values of the four single-qubit $\hat{Z}_i$ observables. As a benchmark, we also compare mitigated expectation values from digital ZNE. In Fig. 2, we show the error (between the mitigated expectation values and the ideal ones) distribution of digital ZNE and ML-QEM with each of the four machine learning models on the top plot and the bootstrap mean errors in the bottom plot. We observe that the random forest consistently outperforms the other ML-QEM models, with the MLP model closely following. Notably, all ML-QEM models, including OLS and GNN, exhibit competitive performance in comparison to the ZNE method, despite that the runtime overhead for ZNE is twice as much. Finally, we emphasize that rigor hyperparameter optimization may impact the relative performance of these methods, and we leave this analysis to future work.

### 2.1.2  Trotterized 1D Transverse-field Ising Model

To benchmark the performance of the protocol on structured circuits, we consider Trotterized brickwork circuits. Here, we consider first-order Trotterized dynamics of the 1D transverse-field Ising model (TFIM) subject to different noise models based on the incoherent noise on the `FakeLima` simulator in Fig. 3, before moving to experiments on IBM hardware with actual device noise in Fig. 4. The dynamics of the spin chain is described by

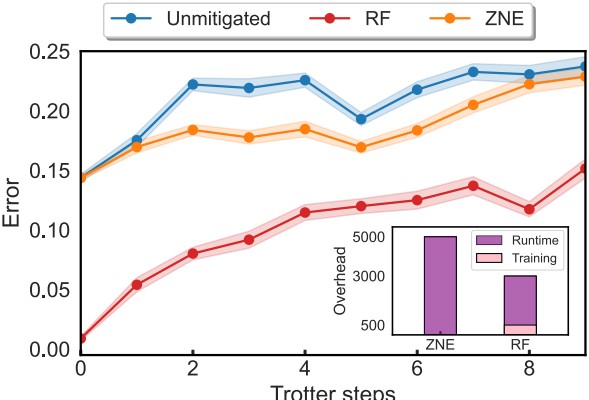

Figure 4: **On QPU hardware: accuracy and overhead for ML-QEM and QEM.** Average execution error of Trotter circuits for experiments on QPU device `ibm_algiers` without mitigation and with ZNE or ML-QEM RF mitigation. Error performance is averaged over $250$ Ising circuits per Trotter step, each with sampled Ising parameters $J < h$ and each measured for all weight-one observables in a randomly chosen Pauli basis. Training is performed over $50$ circuits per Trotter step, which results in both a $40\%$ lower *overall* and $50\%$ lower *runtime* quantum resource overhead of RF compared to the overhead of the digital ZNE implementation (see inset).

the Hamiltonian

$$\hat{H} = -J \sum_j \hat{Z}_j \hat{Z}_{j+1} + h \sum_j \hat{X}_j = -J\hat{H}_{ZZ} + h\hat{H}_X \ ,$$

where $J$ denotes the exchange coupling between neighboring spins and $h$ represents the transverse magnetic field, whose first-order Trotterized circuit is shown in the inset of Fig. 3. We generate multiple instances of the problem with varying numbers of Trotter steps and coupling strengths, such that the coupling strengths of each circuit are uniformly sampled from the paramagnetic phase ($J < h$) by choice. There are $300$ training circuits and $300$ testing circuits per Trotter step, and the training circuits cover Trotter steps up to $14$. Each circuit is measured in a randomly chosen Pauli basis for all the weight-one observables. We then train the ML-QEM models on the ideal and noisy expectation values obtained from these circuits and compare their performance with digital ZNE. During the testing phase, we consider both interpolation and extrapolation. In interpolation, we test on circuits with sampled coupling strength $J$ not included in training but with Trotter steps included in the training. In extrapolation, we test on circuits with sampled coupling strength $J$ not included in the training as well as with Trotter steps exceeding the maximal steps present in the training circuits.

On the noisy simulator in Fig. 3, for this problem with incoherent gate noise in the absence (left) or presence (right) of readout error, the ML-QEM model (using the random forest) performs better than the ZNE method. We present a comparison across all ML-QEM models in App. D, such that the RF model demonstrates the best performance among the ML-QEM models both in interpolation and extrapolation, closely followed by the MLP, OLS, and GNN. We envision that ML-QEM can be used to improve the accuracy of noisy quantum computations for circuits with gate depths exceeding those included in the training set.

On the right of Fig. 3, we consider the same problem in the second study but simulate the sampled circuits on `FakeLima` backend with additional coherent errors. The added coherent errors are CNOT gate over-rotations with an average over-rotational angle of $0.02\pi$. We again generate multiple instances of the problem with varying numbers of Trotter steps and coupling strengths uniformly sampled from the paramagnetic phase.

During the testing phase, the testing circuits cover $14$ more steps up to Trotter step $29$. Under the influence of added coherent noise, the performance of the ML-QEM model and digital ZNE deteriorated compared to the previous study. However, in the extrapolation scenario, none of the models demonstrated effective mitigation of the noisy expectation values. In practical applications, a combination of, e.g., dynamical decoupling and randomized compiling, which can suppress all coherent errors, could be applied to the test circuits prior to utilizing ML-QEM models. This approach effectively converts the noise into incoherent noise, enabling the ML-QEM methods to perform optimally in extrapolation. We remark that coherent gate errors induce quadratic changes in the expectation values, which are stronger than incoherent errors inducing only linear changes—it is plausible that the machine learning approach performs better with weak noises.

We benchmark the performance of the ML-QEM model against digital ZNE on real quantum hardware, IBM's `ibm_algiers`. In this experiment, we do not apply any additional error suppression or error mitigation such as dynamical decoupling, randomized compiling, or readout error mitigation; thus, the experiment involves incoherence noise, coherent noise, and readout error, with the results shown in Fig. 4. We train the ML-QEM with random forest on $50$ circuits and test it on $250$ circuits at each Trotter step. We observe that $50$ training

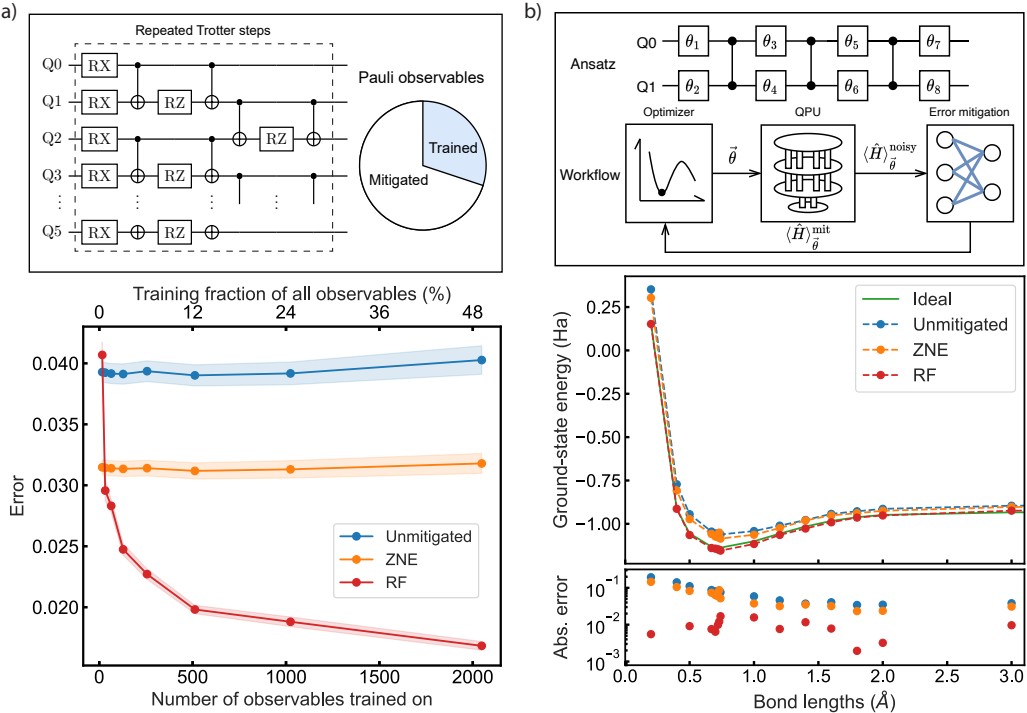

Figure 5: **Application of ML-QEM to a) unseen expectation values and b) the variational quantum eigensolver (VQE).** a) Top: Schematic of a Trotter circuit, which prepares a many-body quantum state on $n = 6$ qubits (in 5 Trotter steps). Top right: Circle depicts the pool of all possible $4^n$ Pauli observables. Shaded region depicts the fraction of observables used in training the ML model; the remaining observables are unseen prior to deployment in mitigation. Bottom: Average error of mitigated unseen Pauli observables versus the total number of distinct observables seen in training. b) Top: Schematic of the VQE ansatz circuit for 2 qubits parametrized by 8 angles $\vec{\theta}$. Below, a depiction of the VQE optimization workflow optimizing the set of angles $\vec{\theta}$ on a simulated QPU, yielding the noisy chemical energy $\langle \hat{H} \rangle_{\vec{\theta}}^{\text{noisy}}$, which is first mitigated by the ML-QEM or QEM before being used in the optimizer as $\langle \hat{H} \rangle_{\vec{\theta}}^{\text{mit}}$. Compared to the ZNE method, the ML-QEM with RF method obviates the need for additional mitigation circuits at every optimization iteration at runtime.

circuits per step, totaling $500$ training circuits, suffices to have the model trained well. With this low train-test split ratio, the ML-QEM requires $500 + 2{,}500 = 3{,}000$ total circuits, while running ZNE with 2 noise factors on the testing circuits requires $2 \times 2{,}500 = 5{,}000$ total circuits. The ML-QEM claims a reduction of quantum resource overhead compared to ZNE *both overall and at runtime*—the reduction is as large as $30\%$ overall and $50\%$ at runtime. Additionally, we observe that the ML-QEM method RF significantly outperforms ZNE for all Trotter steps, demonstrating the efficacy of this approach under a realistic scenario. We report approximately $0.7$ QPU hours (based on a conservative sampling rate of $2$ kHz Kim et al. [2023a]) to generate all the training data and seconds to train the model with a single-core laptop for this experiment.

## 2.2 Mitigating Unseen Pauli Observables

There are algorithms in which we care about the expectation values of multiple non-commuting Pauli observables on the same circuit, effectively creating multiple target circuits with the same gate sequences but with different measuring basis, such as in quantum state tomography and in variational quantum eigensolver. Additional error mitigation methods incur a large overhead on top of these target circuits by requiring additional mitigation circuits for each target circuit. Here, we show that it is possible to achieve better mitigation performance with lower overhead using an ML-QEM method.

In particular, we evaluate the performance of the ML-QEM to mitigate unseen Pauli observables on a state $|\psi\rangle$ produced by the Trotterized Ising circuit depicted on the top of Fig. 5(a), which contains 6 qubits and 5 Trotter steps. We train the random forest model on increasing fractions of the $4^6 - 1 = 4{,}095$ Pauli observables of a Trotterized Ising circuit with $J/h = 0.15$, and then we apply the model to mitigate noisy expectation values sampled from the

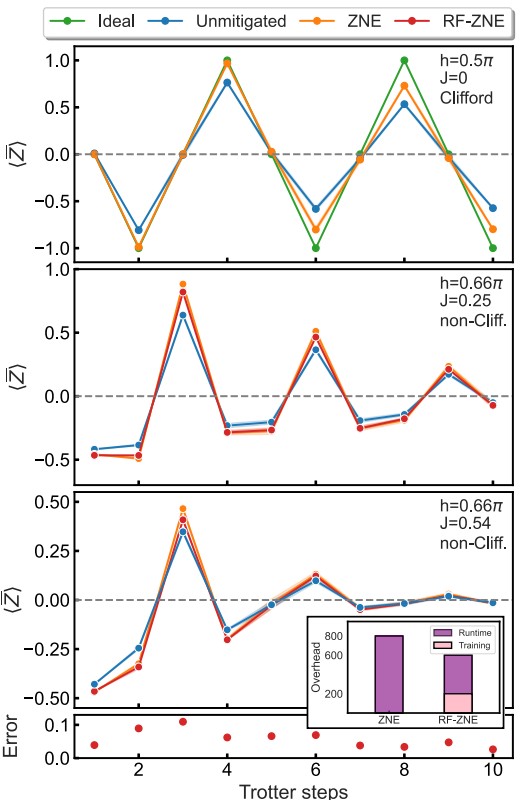

Figure 6: **ML-QEM mimicking QEM on large, 100-qubit circuits with lower overheads, in hardware.** Top three panels: Average expectation values from 100-qubit Trotterized 1D TFIM circuits executed in hardware on QPU `ibm_brisbane`. Each panel corresponds to a different Ising parameter set (top right corners). Top panel corresponds to a Clifford circuit, whose ideal, noise-free expectation values are shown as the green dots. The RF-mimicking-ZNE (RF-ZNE) curve corresponds to training the RF model against ZNE-mitigated data on the hardware rather than in numerical simulators, for which these large non-Clifford circuits are more difficult. This approach enables low-overhead error mitigation when ideal results are not available from classical simulation. Bottom panel: The error, measured again in the $L_2$ norm, between the ZNE-mitigated expectation values and the RF-mimicking-ZNE (RF-ZNE) mitigated expectation values over non-Clifford testing circuits with randomly sampled coupling strengths $J < h$. averaged over 40 testing circuits per Trotter step and the observables. The training is over 10 circuits per Trotter step, which results in a 25% lower *overall* and 50% lower *runtime* quantum resource overhead compared to the ZNE applied in this experiment, as shown in the inset.

*rest* of all possible Pauli observables. The results of this study are plotted at the bottom of Fig. 5(a). We observe that training the random forest on just a small fraction ($\lesssim 2\%$) of the Pauli observables results in mitigated expectation values with errors lower than when using ZNE. The ML-QEM method additionally has lower runtime overhead.

## 2.3 Training on a Variational Ansatz

In the conventional formulation of the variational quantum eigensolver (VQE) algorithm, the goal is to estimate the ground-state energy by measuring the energy $\langle \hat{H} \rangle_{\vec{\theta}}$ of the state prepared by a circuit ansatz $\hat{U}(\vec{\theta})$ with a fixed structure and parameters $\vec{\theta}$. Then, a classical optimizer is used to propose a new $\vec{\theta}$, and this procedure is executed repeatedly until $\langle \hat{H} \rangle_{\vec{\theta}}$ converges to its minimum. When executing this algorithm on a noisy quantum computer, error mitigation can be used to improve the noisy energy $\langle \hat{H} \rangle_{\vec{\theta}}^{\mathrm{noisy}}$ to the mitigated energy $\langle \hat{H} \rangle_{\vec{\theta}}^{\mathrm{mit}}$ and better estimate the ground-state energy. This workflow is shown at the top of Fig. 5(b). Error-mitigated VQE with traditional methods can be costly, however, as additional mitigation circuits must be executed during each iteration. We use ML-QEM error mitigation instead, where a model is trained beforehand to mitigate the ground-state energy of an ansatz $\hat{U}(\vec{\theta})$ so that at each iteration, no additional mitigation circuits need to be executed. A trained model could also then be used for error-mitigated VQE for different Hamiltonians.

To demonstrate this concept, we train the ML-QEM model with RF on 2,000 circuits with each parameter randomly sampled from $[-5, 5]$, and compute the dissociation curve of the $H_2$ molecule on the bottom of Fig. 5(b). The ML-QEM random forest model is trained on a two-local variational ansatz (depicted on the top of Fig. 5(b)) across many randomly sampled $\{\vec{\theta}\}$. This method results in energies that are close to chemical accuracy. Notably, the absolute errors are an order of magnitude smaller than those of the ZNE-mitigated energies.

## 2.4 Scalability through Mimicry

For large-scale circuits whose ideal expectation values of certain observables are inefficient or impossible to obtain by classical simulations, we cannot train the model to mitigate expectation values towards the ideal ones. Rather, we could train the model to mitigate expectation values towards values mitigated by *other* scalable QEM methods, enabling scalability of ML-QEM through mimicry. Mimicry can be concretely visualized using the workflow for ML-QEM depicted in Fig. 1 with an *error-mitigated QPU* selected instead of a *noiseless simulator*, as we show in

the inset of Fig. 6. Performing mimicry does not allow the ML-QEM model to outperform the mimicked QEM method by its nature, but allows the ML-QEM model to reduce the overhead compared to the traditional ML-QEM.

We demonstrate this capability by training an ML-QEM model to mimic digital ZNE in a 100-qubit Trotterized 1D TFIM experiment on `ibm_brisbane`. In particular, we use ZNE to mitigate five single-qubit $\hat{Z}_i$ observables on five qubits on the Ising chain with varying numbers of Trotter steps and $J/h$ values. Each Trotter step contains 4 layers of parallel CNOT gates, and the circuits at Trotter step 10 has 1,500 CNOT gates in total. As shown in the top of Fig. 6, we first confirm that the ZNE-mitigated expectation values are more accurate than the unmitigated ones by benchmarking ZNE on a 100-qubit Trotterized Ising circuit with $h = 0.5\pi$ and $J = 0$ such that it is Clifford and classically simulable. We then train a random forest model to mitigate noisy expectation values the same way that ZNE does. In this experiment, we apply Pauli twirling to all the circuits, each with 5 twirls, before applying either extrapolation in digital ZNE or the ML-QEM to mitigate the expectation values.

We then find that the ML-QEM models are able to accurately mimic the traditional method's mitigated expectation values. The average distance from the unmitigated result (after twirling average) for the mitigated expectation values produced by ZNE and the random forest model mimicking ZNE are very close for all Trotter steps, as shown for specific $J$ and $h$ corresponding to non-Clifford circuits in the second and third plot of Fig. 6. In the fourth and bottom plot showing the residuals between the ZNE-mitigated and RF-mimicking-ZNE-mitigated values averaged over the training set comprising non-Clifford circuits, we see that RF mimics ZNE well. This result demonstrates that ML-QEM methods can scalably accelerate traditional quantum error mitigation methods by mimicking their behavior when exact expectation values cannot be computed classically. In this experiment, although 1D TFIM is analytically solvable, the Trotter errors should be taken into consideration, and thus the exact expectation values of the circuits are not easily accessible, and thus not shown.

Importantly, this mimicry approach requires less quantum computational overhead *both overall and at runtime*. For this experiment, we test on 40 different coupling strengths $J$ for $h = 0.66\pi$, each of which is used to generate 10 circuits with up to 10 Trotter steps, or 400 test circuits in total. The ZNE approach with 2 noise factors requires $2 \times 400 = 800$ circuits. In contrast, the RF-mimicking-ZNE approach here is trained with 10 different coupling strengths $J$ for $h = 0.66\pi$, each of which generates 10 circuits with up to 10 Trotter steps, or 100 total training circuits. Therefore, the RF-mimicking-ZNE approach requires only $2 \times 100 + 400 = 600$ total circuits, resulting in 25% *overall* lower quantum computational resources. The savings are even more drastic *at runtime*—again, the ZNE approach with 2 noise factors requires 2 circuits to be executed per test circuit, whereas each test circuit only has to be executed once for RF-mimicking-ZNE-based mitigation, resulting in 50% savings. We expect the error of the mimicry to shrink should more training data be provided. We report approximately 0.14 QPU hours (based on a conservative sampling rate of 2 kHz Kim et al. [2023a]) to generate all the training data and seconds to train the model with a single-core laptop for this experiment.

## 3 Discussion

In this paper, we have presented a comprehensive study of machine learnin for quantum error mitigation (ML-QEM) methods, including linear regression, random forest, multi-layer perceptrons, and graph neural networks, for improving the accuracy of quantum computations. First, we conducted performance comparisons over many practically relevant contexts; they span circuits (random circuits and Trotterized 1D transverse-field Ising circuits), noise models (qubit decoherence, readout, depolarizing gate, and/or coherent gate errors), and applications (mitigating unseen Pauli observables and enhancing variational quantum eigensolvers) studied here, we find that the best-performing model is the random forest (RF). Second, we demonstrated that ML-QEM methods can perform better than a traditional method, zero-noise extrapolation (ZNE). Paired with the ability to mitigate at runtime by running no additional mitigation circuits, ML-QEM reduces the runtime overhead of traditional methods; for instance, it reduces the runtime overhead by a factor of at least 2 compared to digital ZNE. Therefore, ML-QEM can be especially useful for algorithms where many circuits that are similar to each other are executed repeatedly, such as quantum state tomography-like experiments and variational algorithms. Finally, we find that ML-QEM can even effectively mimic other mitigation methods, providing very similar performance but with a lower overhead at runtime. This allows the ML-QEM to scale up to classically intractable circuits.

## Appendices

## A  Methods

### A.1  Statistical Learning Models

Here, we discuss each of the statistical model (schematics shown in Fig. 7), data encoding strategies, and hyperparameters used in this study. We emphasize that the performance of a model depends on factors such as the size of the training dataset, encoding scheme, model architecture, hyper-parameters, and particular tasks. Therefore, from a broader perspective, we hope that the models in this work provide a sufficient starting point for practitioners of quantum computation with noisy devices to make informed decisions about the most suitable approach for their application.

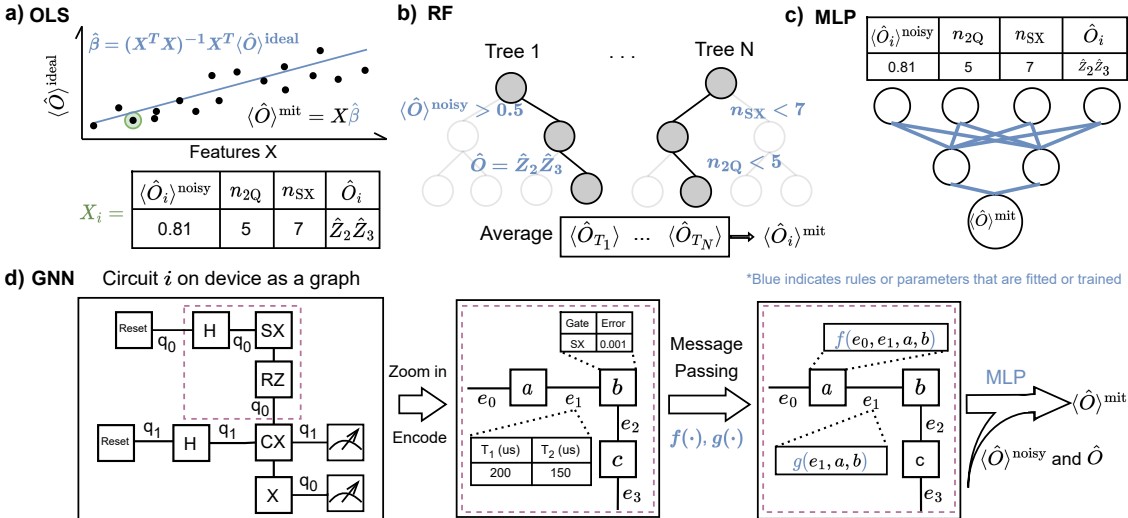

Figure 7: **Overview of the four ML-QEM models and their encoded features.** (a) Linear regression (specifically ordinary least-square (OLS)): input features are vectors including circuit features (such as the number of two-qubit gates $n_{2\mathrm{Q}}$ and SX gates $n_{\mathrm{SX}}$), noisy expectation values $\langle\hat{O}\rangle^{\mathrm{noisy}}$, and observables $\hat{O}$. The model consists of a linear function that maps input features to mitigated values $\langle\hat{O}\rangle^{\mathrm{mit}}$. (b) Random forest (RF): the model consists of an ensemble of decision trees and produces a prediction by averaging the predictions from each tree. (c) Multi-layer perception (MLP): the same encoding as that for linear regression is used, and the model consists of one or more fully connected layers of neurons. The non-linear activation functions enable the approximation of non-linear relationships. (d) Graph neural network (GNN): graph-structured input data is used, with node and edge features encoding quantum circuit and noise information. The model consists of multiple layers of message-passing operations, capturing both local and global information within the graph and enabling intricate relationships to be modeled.

### A.1.1  Linear Regression

Linear regression is a simple and interpretable method for ML-QEM, where the relationship between dependent variables (the ideal expectation values) and independent variables (the features extracted from quantum circuits and the noisy expectation values) is modeled using a linear function.

One relevant work in this area is Clifford data regression, proposed by Czarnik et al. Czarnik et al. [2021]. In their approach, the authors first replace most of the non-Clifford gates with nearby Clifford gates in the target circuit of interest, then use a linear regression model to regress the noisy expectation values of those circuits onto the ideal ones. Our linear regression model differs in two main aspects. Firstly, we extend the feature set to include counts of each native gate where native parameterized gates are counted in binned angles, the Pauli observable in sparse Pauli operator representation, and optional device-specific noise parameters. Secondly, our model does not necessarily require training on Clifford versions of the target circuits, although this option remains available if desired.

We train a linear regression model that takes these features as input and predicts the ideal expectation values. The model minimizes the sum squared error between the mitigated and the ideal expectation values using a closed-form

solution, which is named ordinary least squares (OLS). The linear regression model can also be trained using other methods, such as ridge regression, LASSO, or elastic net. These methods differ in their regularization techniques, which can help prevent overfitting and improve model generalization. In our experiments, we use OLS for its simplicity and ease of interpretation. We note that standard feature selection procedures also help to prevent overfitting and collinearity in practice.

### A.1.2 Random Forest

Random forest (RF) is a robust, interpretable, non-linear decision tree-based model to perform quantum error mitigation. As an ensemble learning method, it employs bootstrap aggregating to combine the results produced from many decision trees, which enhances prediction accuracy and mitigates overfitting. Moreover, each decision tree within the random forest utilizes a random subset of features to minimize correlation between trees, further improving prediction accuracy.

The input features to the random forest model are extracted from the quantum circuits, specifically counts of each native gate on the backend (native parameterized gates are counted in binned angles), the Pauli observable in sparse Pauli operator representation, and optional device-specific noise parameters. We train a random forest regressor with a specified large number of decision trees on the training data. Given all the features, the random forest model averages the predictions from all its decision trees to produce an estimate of the ideal expectation value.

For RF, we used 100 tree estimators for each observable. The tree construction process follows a top-down, recursive, and greedy approach, using the Classification and Regression Trees (CART) algorithm. For the splitting criterion, we employ the mean squared error reduction for regressions. For each tree, at least 2 samples are required to split an internal node, and 1 feature is considered when looking for the best split.

### A.1.3 Multi-Layer Perceptron

Multi-layer perceptrons (MLPs), first explored in the context of QEM in Ref. Kim et al. [2020], are feedforward artificial neural networks composed of layers of nodes, with each layer fully connected to the subsequent one. Nodes within the hidden layers utilize non-linear activation functions, such as the rectified linear unit (ReLU), enabling the MLP to model non-linear relationships.

We construct MLPs with 2 dense layers with a hidden size of 64 and the ReLU activation function. The input features are identical to those employed in the random forest model. To train the MLP, we minimize the mean squared error between the predicted and true ideal expectation values, employing backpropagation to update the neurons. The batch size is 32, and the optimizer used is Adam Kingma and Ba [2015] with an initial learning rate of 0.001. In practice, regularization techniques like dropout or weight decay can be used to prevent overfitting if necessary. The MLP method demonstrates competitive performance in mitigating noisy expectation values, as evidenced by our experiments. However, it should be noted that MLPs are also susceptible to overfitting in this context.

### A.1.4 Graph Neural Network

As the most complex model among the four, graph neural networks (GNNs) are designed to work with graph-structured data, such as social networks Ying et al. [2018] and chemistry Reiser et al. [2022]. They can capture both local and global information within a graph, making them highly expressive and capable of modeling intricate relationships. However, their increased complexity results in higher computational costs, and they may be more challenging to implement and interpret.

A core aspect of our ML-QEM with GNN lies in data encoding, which consists of encoding quantum circuits, and device noise parameters into graph structures suitable for GNNs. Before data encoding, each quantum circuit is first transpiled into hardware-native gates that adhere to the quantum device's connectivity. To encode them for GNN, the transpiled circuit is converted into an acyclic graph. In the graph, each edge signifies a qubit that receives instructions when directed towards a node, while each node corresponds to a gate. These nodes are assigned vectors containing information about the gate type, gate errors, as well as the coherence times and readout errors of the qubits on which the gate operates. Additional device and qubit characterizations, such as qubit crosstalk and idling period duration, can be encoded on the edge or node, although they are not considered in the current study.

The acyclic graph of a quantum circuit, serves as input to the transformer convolution layers of the GNN. These message-passing layers iteratively process and aggregate encoded vectors on neighboring nodes and connected

edges to update the assigned vector on each node. This enables the exchange of information based on graph connectivity, facilitating the extraction of useful information from the nodes which are the gate sequence in our context. The output, along with the noisy expectation values, is passed through dense layers to generate a graph-level prediction, specifically the mitigated expectation values. As a result, after training the layers using backpropagation to minimize the mean squared error between the noisy and ideal expectation values, the GNN model learns to perform quantum error mitigation.

For the GNN, we use 2 multi-head Transformer convolution layers Shi et al. [2021] and ASAPooling layers Ranjan et al. [2019] followed by 2 dense layers with a hidden size of 128. Dropouts are added to various layers. As with the MLP, the batch size is 32, and the optimizer used is Adam Kingma and Ba [2015] with an initial learning rate of 0.001.

## A.2 Zero-Noise Extrapolation

We use zero-noise extrapolation with digital gate folding on 2-qubit gates, noise factors of $\{1, 3\}$, and linear extrapolation implemented via Ref. Rivero et al. [2022].

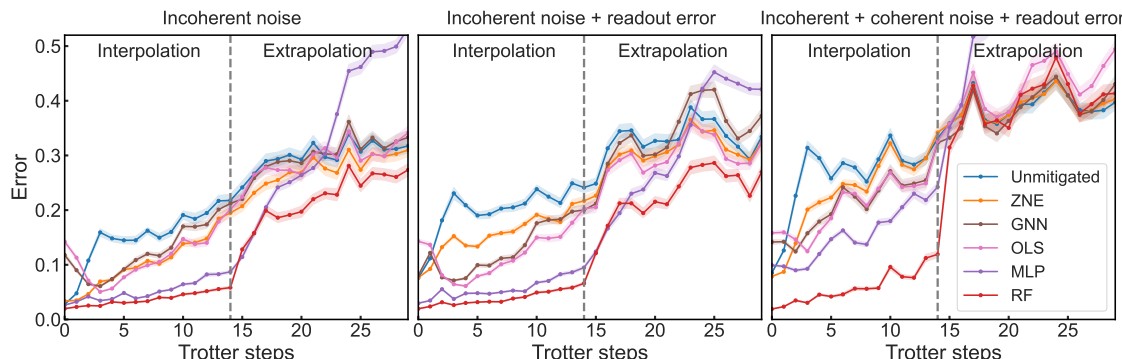

Figure 8: **ML-QEM and QEM performance for Trotter circuits.** Expanded data corresponding to Fig. 3 of the main text that includes the three ML-QEM methods not shown earlier: GNN, OLS, MLP. We study three noise models: Left: incoherent noise resembling `ibmq_lima` without readout error, Middle: with the additional readout error, and Right: with the addition of coherent errors on the two-qubit CNOT gates. We show the depth-dependent performance of error mitigation averaged over 9,000 Ising circuits, each with different coupling strengths $J$. For the incoherent noise model, all ML-QEM methods demonstrate improved performance even when mitigating circuits with depths larger than those included in the training set. However, all perform as poorly as the unmitigated case in extrapolation with additional coherent noise.

## B   Depolarizing Noise

We show here that the ideal expectation values of an observable $\hat{O}$ linearly depend on its noisy expectation values when the noisy channel of the circuit consists of successive layers of depolarizing channels. This is more general than the result shown in Czarnik et al. [2021].

Consider $l$ successive layers of unitaries each associated with a depolarizing channel with some rate $p_l$, the noisy circuit acting on the input $\rho$, $\tilde{\mathcal{C}}(\rho)$, is written as $\tilde{\mathcal{C}}(\rho) = \mathcal{E}_l(U_l \mathcal{E}_{l-1}(U_{l-1} \ldots \mathcal{E}_1(U_1 \rho U_1^\dagger) \ldots U_{l-1}^\dagger) U_l^\dagger)$, where $\mathcal{E}_l(\rho) = (p_l/D)I + (1 - p_l)\rho$.

It can be shown by induction that $\tilde{\mathcal{C}}(\rho) = (p(l)/D)I + (1-p(l))U_l \ldots U_1 \rho U_1^\dagger \ldots U_l^\dagger$, where $p(l) = 1 - \Pi_{i=1}^l (1-p_i)$ as follows. Assuming for $l = k$, $\tilde{\mathcal{C}}(\rho) = (p(k)/D)I + (1 - p(k))U_k \ldots U_1 \rho U_1^\dagger \ldots U_k^\dagger$, then for $l = k + 1$, we have $\tilde{\mathcal{C}}(\rho) = (p(k)/D)I + (1 - p(k))[p_{k+1}I/D + (1 - p_{k+1})U_k \ldots U_1 \rho U_1^\dagger \ldots U_k^\dagger] = (p(k+1)/D)I + (1 - p(k+1))U_k \ldots U_1 \rho U_1^\dagger \ldots U_k^\dagger$. The induction completes with a trivial base case.

Therefore, the noisy expectation value of $\hat{O}$ becomes

$$\text{Tr}(\tilde{\mathcal{C}}(\rho)\hat{O})$$
$$= \frac{p(l)}{D}\text{Tr}(\hat{O}) + (1 - p(l))\text{Tr}(U_l \ldots U_1 \rho U_1^\dagger \ldots U_l^\dagger \hat{O})$$
$$= \frac{p(l)}{D}\text{Tr}(\hat{O}) + (1 - p(l))\text{Tr}(\mathcal{C}(\rho)\hat{O}) \, ,$$

where $\text{Tr}(\mathcal{C}(\rho)\hat{O})$ is the ideal expectation value of $\hat{O}$.

For Trotterized circuits with a fixed Trotter step and a fixed brickwork structure, the number of layers $l$ of unitaries in the circuit is also fixed. Assuming some fixed-rate depolarizing channels associated with the $l$ layers of unitaries, the noisy and ideal expectation values of some $\hat{O}$ on these Trotterized circuits with different parameters then lie on a line. Therefore, the ML-QEM method can mitigate the expectation values by linear regression from the noisy expectation values to the ideal ones, and the linear regression parameters can be learned to vary according to the number of layers $l$. The ML-QEM is thus *unbiased* in this case. We note that ZNE with linear extrapolation is still *biased* in this case, since the noise amplification effectively results in a different combined depolarizing rate $p'(l) = 1 - \Pi_{i=1}^{l}(1 - p_i')$, which leads to expectation values with differently amplified noises each lying on a different line towards the ideal expectation value, and thus the linear extrapolation cannot yield unbiased estimates.

## C   Break-even in the total cost of the ML-QEM

Assuming the mimicked QEM requires $m$ total executions of either the mitigation circuits or the circuit of interest (e.g., digital/analog ZNE usually has $m = 2$ or 3 noise factors), the total cost of the mimicked QEM, namely its runtime cost, is $m n_{\text{test}}$. The total cost, including training, for the RF is $m n_{\text{train}} + n_{\text{test}}$. Equating these two yields the break-even train-test split ratio in the total cost of our mimicry compared to the traditional QEM: $n_{\text{train}}/n_{\text{test}} = (m - 1)/m$. Our mimicry shows a higher overall efficiency when the train-test split ratio is smaller than $(m - 1)/m$.

## D   Additional Experimental Details

All non-ideal expectation values in simulations and experiments presented in this paper are obtained from the measurement statistics from 10,000 shots.

In the study of 4-qubit random circuits presented in Sec. 2.1.1, to generate the random circuits, we use the Qiskit function `qiskit.circuit.random.random_circuit()`, which implements random sampling and placement of 1-qubit and 2-qubit gates, with randomly sampled parameters for any selected parametrized gates. The 2-qubit gate depth is measured after transpilation. We remark that the random circuits sampled at large depths may approximate the Haar distribution and have expectation values concentrated around 0 to some extent Harrow and Low [2009], Liao et al. [2021].

In the study of the Trotterized 1D TFIM in Sec. 2.1.2, we initialize the state devoid of spatial symmetries. This is done to intentionally introduce asymmetry in the single-qubit $\hat{Z}_i$ expectation value trajectories across Trotter steps, thereby increasing the difficulty of the regression task. Conversely, when the initial state possesses a certain degree of symmetry, the regression analysis, which incorporates noisy expectation values as features, becomes highly linear, resulting in a strong performance by the OLS method.

We present a comparison across all ML-QEM models in the study of mitigating expectation values of Trotterized 1D TFIM in Fig. 8. With incoherent noise only, the random forest model demonstrates the best performance among the ML-QEM models both in interpolation and extrapolation, closely followed by the MLP, OLS, and GNN. With additional coherent noise, in the interpolation scenario, the performance ranking of the other models remained largely consistent with that observed in the previous study. Notably, the random forest model exhibited the best performance among the ML-QEM models, closely followed by the MLP model.

We observe that both in the simulation and in the experiment of the small-scale Trotterized 1D TFIM, there are significant correlations between the noisy expectation values and the ideal ones. There are also significant correlations but to a lesser degree between the gate counts and the ideal expectation values, suggesting the models are using certain depth information deduced from the gate counts to correct the noisy expectation values towards the ideal ones.

## E   Efficient training and fine-tuning

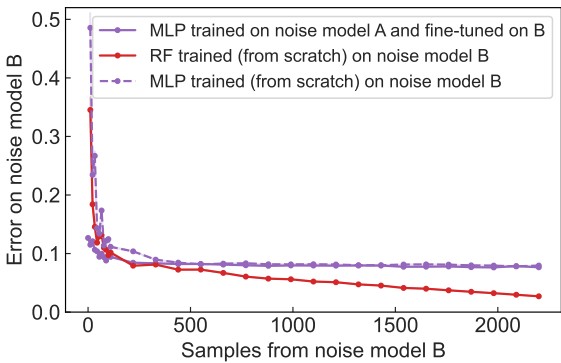

Figure 9: **Updating the ML-QEM models on the fly.** Comparing the efficiency and performance of ML models, fine-tuned or trained from scratch, on a different noise model. Noise model A represents `FakeLima` and noise model B represents `FakeBelem`. All training, fine-tuning, and testing circuits are 4-qubit 1D TFIM measured in a random Pauli basis for four weight-one observables. The solid purple curve shows the testing error on noise model B of an MLP model originally trained on 2,200 circuits run on noise model A and fine-tuned incrementally with circuits run on noise model B. The dashed purple curve shows the testing error on noise model B of another MLP model trained only on circuits from noise model B. The red curve shows the testing error on noise model B of an RF model trained only on circuits from noise model B. All three methods converge with a small number of training/fine-tuning samples from noise model B. While the testing error of the fine-tuned and trained-from-scratch MLP models converged, both were outperformed by a trained-from-scratch RF model. This provides evidence that ML-QEM can be efficient in training.

Because the noise in quantum hardware can drift over time, an ML-QEM model trained on circuits run on a device at one point in time may not perform well at another point in time and may require adaption to the drifted noise model on the device. Therefore, we explore whether an ML-QEM model can be fine-tuned for a different noise model and show that similar performance can be achieved with substantially less training data.

In particular, we fine-tune an MLP and compare its learning rate against RF. The MLP can be fine-tuned on a different noise model after they have been originally trained on a noise model. The fine-tuning is expected to require only a small number of additional samples—this is demonstrated in Fig. 9 with the MLP trained on noise model A (`FakeLima`) and fine-tuned on noise model B (`FakeBelem`) which converges after $\sim 300$ fine-tuning circuits. On the other hand, an MLP trained from scratch and tested on a noise model B shows a slower convergence after $\sim 500$ training circuits, though both fine-tuning and training from scratch produce the same testing performance. We also compare them with an RF trained from scratch, which converges after fewer than $\sim 300$ training circuits, demonstrating the excellent efficiency in training an RF model. While future research can investigate in more detail the drift in noise affecting the ML model performance, we show evidence that MLP can be efficiently adapted to new device noise and that RF can be trained as efficiently from scratch to new device noise.

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
