# OpenReview forum: "Machine Learning for Practical Quantum Error Mitigation"
_NeurIPS.cc/2023/Workshop/AI4Science — NeurIPS2023-AI4Science Poster_

### Official Review · Reviewer_rygL · 2023-10-20

**Rating:** 8
**Confidence:** 3

**Review:**

This paper conducts both simulations and experiments on state-of-the-art quantum computers using up to 100 qubits, we demonstrate that machine learning for quantum error mitigation (ML-QEM) can drastically reduce overheads, maintain or even surpass the accuracy of conventional methods, and yield near noise-free results for quantum algorithms. These results could make profound impact for the society of science.